# OGR1 (GPR68) and TDAG8 (GPR65) Have Antagonistic Effects in Models of Colonic Inflammation

**DOI:** 10.3390/ijms241914855

**Published:** 2023-10-03

**Authors:** Leonie Perren, Moana Busch, Cordelia Schuler, Pedro A. Ruiz, Federica Foti, Nathalie Weibel, Cheryl de Vallière, Yasser Morsy, Klaus Seuwen, Martin Hausmann, Gerhard Rogler

**Affiliations:** Department of Gastroenterology and Hepatology, University Hospital Zurich, University of Zurich, 8091 Zurich, Switzerland; leonie.perren@usz.ch (L.P.); moana.busch-dohr@usz.ch (M.B.); cordelia.schuler@gmx.ch (C.S.); pa.ruizcastro@gmail.com (P.A.R.); federica.foti@uzh.ch (F.F.); nathalie.weibel@uzh.ch (N.W.); cheryl.devalliere@usz.ch (C.d.V.); yasser.morsy@usz.ch (Y.M.); klaus.seuwen@sfr.fr (K.S.); gerhard.rogler@usz.ch (G.R.)

**Keywords:** Crohn’s disease, inflammatory bowel disease, OGR1/GPR68, pH-sensing G-protein-coupled receptors, TDAG8/GPR65, ulcerative colitis

## Abstract

G-protein-coupled receptors (GPRs), including pro-inflammatory ovarian cancer GPR1 (OGR1/GPR68) and anti-inflammatory T cell death-associated gene 8 (TDAG8/GPR65), are involved in pH sensing and linked to inflammatory bowel disease (IBD). OGR1 and TDAG8 show opposite effects. To determine which effect is predominant or physiologically more relevant, we deleted both receptors in models of intestinal inflammation. Combined *Ogr1* and *Tdag8* deficiency was assessed in spontaneous and acute murine colitis models. Disease severity was assessed using clinical scores. Colon samples were analyzed using quantitative polymerase chain reaction (qPCR) and flow cytometry (FACS). In acute colitis, *Ogr1*-deficient mice showed significantly decreased clinical scores compared with wildtype (WT) mice, while *Tdag8*-deficient mice and double knockout (KO) mice presented similar scores to WT. In *Il-10-*spontaneous colitis, *Ogr1*-deficient mice presented significantly decreased, and *Tdag8*-deficient mice had increased inflammation. In the *Il10*^−/−^ ×* Ogr1*^−/−^ × *Tdag8*^−/−^ triple KO mice, inflammation was significantly decreased compared with *Tdag8*^−/−^. Absence of *Ogr1* reduced pro-inflammatory cytokines in *Tdag8*-deficient mice. *Tdag8*^−/−^ had significantly more IFNγ^+^ T-lymphocytes and IL-23 T-helper cells in the colon compared with WT. The absence of OGR1 significantly alleviates the intestinal damage mediated by the lack of functional TDAG8. Both OGR1 and TDAG8 represent potential new targets for therapeutic intervention.

## 1. Introduction

OGR1 and TDAG8 are pH-sensing GPRs that belong to the GPR4 subfamily. They are activated by acidic extracellular pH [1,2,3], reaching their maximal activation at approximately pH 6.8, and are silent at pH higher than 7.6 [1,4,5,6]. TDAG8 and OGR1 sense extracellular protons through histidine residues located in the extracellular region of the receptors, resulting in signaling pathway activation and modification of a variety of cell functions. Intracellularly, TDAG8 and OGR1 couple to different Gα protein subunits, triggering the activation of their corresponding downstream signaling pathways [7].

TDAG8 is present in immune cell-specific cells [8] and is found on T cells, macrophages, eosinophils, mast cells, and other cells of the innate immune system [8,9,10,11]. The presence of TDAG8 protects from intestinal inflammation as *Tdag8* deficiency exacerbates intestinal inflammation and fibrosis in the dextran sodium sulfate (DSS)-induced chronic colitis [12], the DSS-induced acute colitis [13], and the T cell transfer-induced colitis mouse models [13]. TDAG8 can be activated by acidic extracellular pH through the protonation of several histidine residues on the receptor extracellular domains and activate downstream signals through the Gα_s_/cAMP [14] and Gα_12/13_/Rho pathways [15]. TDAG8 was proposed to act as a negative regulator of inflammation [13,16] through the activation of a Gα_s_-coupled mechanism [5]. TDAG8 has been identified as an IBD risk gene in genome-wide association studies [17]. In addition to IBD, TDAG8 has been identified as a risk gene for other inflammatory diseases, such as COPD, asthma, multiple sclerosis, and ankylosing spondylitis [18,19,20,21].

OGR1 is expressed in various cell types including intestinal macrophages, granulocytes, endothelial cells, and fibroblasts [22]. The presence of OGR1 perpetuates intestinal inflammation as *Ogr1* deficiency protects from fibrosis [22] and intestinal inflammation in the spontaneous colitis mouse model [23]. Confirmatively, the positive effect of an *Ogr1* inhibitor in the acute and chronic DSS colitis mouse models has been demonstrated [24]. OGR1 couples predominantly through G_q11_ proteins, leading to activation of the phospholipase C (PLC)/inositol phosphate (IP)/Ca^2+^/extracellular signal-regulated kinases (ERK) pathway [1,2] and the Gα_12/13_/Rho signaling pathway [8,25,26,27]. OGR1 activation has a pro-inflammatory effect in colitis by increasing immune and inflammatory factors, as well as genes associated with cell adhesion, ECM remodeling, fibrosis, and organization of actin cytoskeleton among other genes, resulting in elevated fibrosis and a more severe inflammation [23,24,28]. The intestine of IBD patients shows increased mRNA expression of OGR1, especially upon inflammation, compared with the intestine of healthy individuals [22]. Furthermore, the mRNA expression of OGR1 correlates with clinical scores given to IBD patients, indicating that OGR1 has a clinically relevant pro-inflammatory effect that could constitute a therapeutic target [22,23]. In mice, increased OGR1 mRNA expression has also been shown in colon tumors when compared with normal colon samples, confirming a role for OGR1 in both colitis and tumorigenesis [29].

TDAG8 and OGR1 appear to have antagonistic effects, also referred to as “OGR1-TDAG8 reciprocity” [7]. Ogr1 deficiency is mostly anti-inflammatory and therefore protective, whilst Tdag8 deficiency causes opposite effects. To date, there are no data on a potential mutual influence of the pathways activated by both receptors, and their opposite effects on inflammation seem to arise from the activation of different transcription factors and cellular processes. Thus, OGR1 appears to block autophagy and induce the activation of the endoplasmic reticulum stress transcription factor X-box-binding protein-1 through the IRE1α-JNK signaling pathway [30]. Following activation by an acidic environment, different downstream signals from both receptors seem to trigger independent molecular mechanisms whose contribution ultimately modulates inflammation under acidic conditions. The combined effect of both deficiencies has not yet been determined. 

Studies of recent years have shown that GPRs, which are involved in the pH sensing of the intercellular space, are linked to IBD [1,23,29,31,32]. IBD consists of two main phenotypes: ulcerative colitis (UC) and CD. In 2017, there were 6.8 million cases of IBD globally [33]. IBD is characterized by a chronic inflammation of the intestinal wall leading to severe and persistent mucosal damage. An increasing body of studies have provided evidence that the pathogenesis of IBD is associated with genetic susceptibility [17], intestinal microbiota [34], environmental factors, and immunological abnormalities [35]. Strikingly, low pH accompanies the course of the disease: local acidification in the gut lumen has been observed during intestinal inflammation and is implicated in the pathogenesis and progression of IBD [36,37]. Fecal fluid of patients suffering from severe UC was characterized by low fecal pH, bicarbonate, and very high lactate levels [38]. In addition, low pH occurs in chronic inflammation and inflammation-induced tumors due to hypoxia, low perfusion, and production of glycolytic metabolites [39]. This is the first study where the physiological consequences of the double deletion of OGR1 and TDAG8 have been investigated. The aim of this study was to better understand the link between the physiological role of both receptors and assess which one has a predominant effect or is physiologically more relevant in the context of intestinal inflammation. We hypothesized that deletion of pro-inflammatory Ogr1 balances out the absence of anti-inflammatory Il-10 and Tdag8, and consequently OGR1 inhibition represents a potential therapeutic option in IBD, particularly for patients carrying mutations in the CD susceptibility gene TDAG8.

## 2. Results

### 2.1. Ogr1 Deficiency Reduces Clinical Severity in the DSS-Induced Acute Colitis Model, Both in a WT and Tdag8^−/−^ Background

To investigate the interplay of *Ogr1* and *Tdag8* in the development of acute intestinal inflammation, 12–15-week-old WT (*n* = 21), *Ogr1*^−/−^ (*n* = 10), *Tdag8*^−/−^ (*n* = 14), and *Ogr1*^−/−^ × *Tdag8*^−/−^ (*n* = 10) littermate mice were exposed to DSS for 8 days, which resulted in pronounced colitis. As a control, WT mice (*n* = 7) and *Ogr1^−/−^* × *Tdag8*^−/−^ mice (*n* = 5) were exposed to water (H_2_O) for the same duration. As observed earlier, *Tdag8*^−/−^ deficiency did not lead to further body weight loss or a further deteriorated disease activity score compared with WT mice upon DSS. However, *Ogr1*-deficient mice upon DSS showed significantly less body weight loss (Figure 1A) and a significantly decreased disease activity score (Figure 1B) compared with the corresponding WT (*p* = 0.0026 and *p* = 0.0148) and *Tdag8^−/−^* genotypes (*p* = 0.0031 and *p* = 0.0144), respectively. The H_2_O control groups showed no differences between WT and double KO mice (*p* = 0.8687). Before euthanasia, an endoscopy was performed on all mice (Figure 1C). The resulting murine endoscopic index of colitis severity [MEICS] score (Figure 1D) showed significantly decreased scores for *Ogr1-*deficient mice, compared with WT (*p* < 0.0055) and *Tdag8*^−/−^ mice (*p* = 0.0064) upon DSS. Additionally, the scores of *Ogr1*^−/−^ × *Tdag8*^−/−^ mice were significantly decreased when compared with WT (*p* = 0.0033) and *Tdag8*^−/−^ (*p* = 0.0024) mice upon DSS (Figure 1D). A score for surface, vascularity, and transparency, used to calculate the MEICS, were also assessed separately (Appendix A). The surface score was significantly decreased in the *Ogr1*^−/−^ mice compared with WT mice (*p* < 0.001) and *Tdag8*^−/−^ mice (*p* < 0.001) upon DSS. Both the vascularity and the transparency score were significantly decreased in the *Ogr1*^−/−^ mice compared with WT mice (*p* = 0.0110 and 0.0116, respectively) upon DSS. 

The relative spleen weight measured postmortem (Figure 1E) was significantly decreased in *Ogr1^−/−^* mice upon DSS compared with WT mice (*p* < 0.0005) and showed no significant difference compared with the H_2_O control mice, indicating less leukocyte accumulation (Figure 1E). Similarly, the *Ogr1*^−/−^ × *Tdag8*^−/−^ mice upon DSS showed significantly decreased relative spleen weight compared with the *Tdag8*^−/−^ (*p* = 0.0302) mice upon DSS (Figure 1E). *Ogr1^−/−^* mice upon DSS showed a significantly greater colon length when compared with WT mice upon DSS (*p* < 0.0114) (Figure 1F). The other groups exposed to DSS showed a decrease in relative colon length similar to the H_2_O control mice. 

Histology samples were used to assess the severity of inflammation in colon specimens from all groups using HE staining (Figure 2A). The histological score (Figure 2B) was significantly lower in *Ogr1^−/−^* mice with DSS-induced acute colitis compared with the other DSS groups (WT [*p* = 0.001] and *Tdag8*^−/−^ [*p* < 0.0024]). The histological score of *Ogr1*-deficient mice showed no significant differences when compared with that of the WT H_2_O control mice (Figure 2B). The *Ogr1*^−/−^ × *Tdag8*^−/−^ mice upon DSS showed a decreased histological score compared with the *Tdag8*^−/−^ and WT mice upon DSS, but statistical significance was not reached (*p* = 0.5115, both). Lymphocyte infiltration and epithelial damage, used to calculate the histology score, were also assessed separately (Appendix A). The lymphocyte infiltration score was significantly decreased in the *Ogr1*^−/−^ mice compared with WT mice (*p* = 0.0028) and *Tdag8*^−/−^ mice (*p* < 0.001) upon DSS. In contrast, lymphocyte infiltration remained unchanged in *Tdag8*^−/−^ mice compared with WT mice upon DSS (*p* = 0.7228). In the *Ogr1*^−/−^ × *Tdag8*^−/−^ double KO mice, lymphocyte infiltration was significantly decreased compared with *Tdag8*^−/−^ mice (*p* = 0.0349), indicating that the absence of *Ogr1* can also improve inflammation in the context of a *Tdag8* deficiency. The epithelial damage score was significantly decreased in *Ogr1*^−/−^ mice compared with WT (*p* < 0.001), *Tdag8*^−/−^ (*p* = 0.0091), or *Ogr1*^−/−^ × *Tdag8*^−/−^ mice (*p* = 0.0117) upon DSS. In contrast to the lymphocyte infiltration sub-analysis, the epithelial damage score showed no amelioration for the *Ogr1*^−/−^ × *Tdag8*^−/−^ double KO mice compared with *Tdag8*^−/−^ mice or WT mice upon DSS.

### 2.2. Il-10 Deficiency: The Absence of Tdag8 Exacerbates Colitis, While the Absence of Ogr1 Is Protective

In order to extend our studies, the *Il-10^−/−^* spontaneous colitis mouse model was applied. For that, *Il-10^−/−^* × *Ogr1^−/−^* (*n* = 13), *Il-10^−/−^* × *Tdag8*^−/−^ (*n* = 8), and *Il-10*^−/−^ × *Ogr1*^−/−^ × *Tdag8*^−/−^ (*n* = 9) mice were investigated. As a control, littermates from the respective KO with *Il-10*-deficiency were used (total, *n* = 17; *Il-10^−/−^* × *Ogr1^−/−^* littermates, *n* = 9; *Il-10^−/−^* × *Tdag8*^−/−^ littermates, *n* = 5; and *Il-10*^−/−^ × *Ogr1*^−/−^ × *Tdag8*^−/−^ littermates, *n* = 3). Successful induction of colitis was confirmed by an overall body weight loss (Figure 3A). Of note, and in contrast to the DSS model, *Tdag8* deficiency in the *Il10*^−/−^ background led to a clearly exacerbated disease, as demonstrated by a significant increase in body weight loss (*p* = 0.0107). Loss of OGR1 was protective: *Il-10^−/−^* × *Ogr1^−/−^* mice showed a significantly higher body weight at euthanasia compared with *Il-10*^−/−^ (*p* = 0.0275), *Il-10^−/−^* × *Tdag8*^−/−^ (*p* = 0.0003), and *Il-10*^−/−^ × *Ogr1*^−/−^ × *Tdag8*^−/−^ mice (*p* = 0.0339) (Figure 3A). The disease activity score was obtained daily (Figure 3B) and showed a significant decrease at euthanasia for mice of the *Il-10*^−/−^ × *Ogr1*^−/−^ group compared with *Il-10^−/−^* mice (*p* = 0.0002) or *Il-10^−/−^* × *Tdag8*^−/−^ mice (*p* < 0.0001). Before euthanasia, an endoscopy was performed on all mice (Figure 3C). The resulting MEICS score showed that *Il-10^−/−^* × *Ogr1^−/−^* mice presented an ameliorated colitis compared with their *Il-10^−/−^* littermates (*p* < 0.0001, Figure 3D). In contrast, *Il-10^−/−^* × *Tdag8^−/−^* mice showed a significant higher MEICS score compared with their *Il-10^−/−^* littermates (*p* < 0.0001, Figure 3D). Triple KO mice showed a significantly lower MEICS score as compared with *Il-10^−/−^* (*p* = 0.0173, Figure 3D). *Il-10^−/−^* × *Tdag8*^−/−^ mice showed a significant increase in spleen weight (Figure 3E) compared with other groups (*Il-10^−/−^* × *Ogr1^−/−^* [*p* < 0.0001], and *Il-10*^−/−^ × *Ogr1*^−/−^ × *Tdag8*^−/−^ [*p* = 0.0221]). 

On the other hand, *Il-10^−/−^* × *Ogr1^−/−^* mice showed a significantly lower spleen weight compared with *Il-10^−/−^* animals (*p* = 0.0057). There was a reduction in spleen weight when *Il-10^−/−^* × *Tdag8*^−/−^ mice were compared with *Il-10*^−/−^ × *Ogr1*^−/−^ × *Tdag8*^−/−^ mice (*p* = 0.0221). *Il-10*^−/−^ × *Ogr1*^−/−^ mice showed a significantly greater colon length when compared with *Il-10*^−/−^ (*p* < 0.0007, Figure 3F). *Il-10*^−/−^ × *Ogr1*^−/−^ × *Tdag8*^−/−^ mice showed a significantly greater colon length when compared with *Il-10*^−/−^ (*p* = 0.0446) or *Il-10*^−/−^ × *Tdag8*^−/−^ mice (*p* < 0.0112). 

To further assess the severity of inflammation, the distal 1 cm of the colon from all the specimens were subjected to histology. The histological score of the colon (Figure 4A,B and Appendix A) was significantly decreased in *Il-10^−/−^* × *Ogr1^−/−^* mice compared with *Il-10^−/−^* mice (*p* < 0.0001), confirming the protective nature of an *Ogr1* deficiency. As expected, *Il-10^−/−^* × *Tdag8*^−/−^ mice showed a significant exacerbation of the disease compared with *Il-10^−/−^* mice (*p* < 0.0001), including cryptitis (Appendix A). Interestingly, triple KO mice showed no significant difference compared with *Il-10^−/−^* × *Ogr1^−/−^* mice (*p* = 0.1601) and *Il-10^−/−^* mice (*p* = 0.2666), but a significantly lower score compared with *Il-10^−/−^* × *Tdag8*^−/−^ mice (*p* < 0.0001). These results demonstrate that the protective effect of the *Ogr1* depletion is maintained in a *Tdag8*^−/−^ background. *Il-10^−/−^* × *Tdag8*^−/−^ mice showed a significantly higher histological score compared with all other groups (*p* < 0.0001).

Additionally, a histological score of the small bowel was performed (Appendix A). The results support the findings from the histological score of the colon and showed a significantly decreased score in *Il-10^−/−^* × *Ogr1^−/−^* mice compared with *Il-10^−/−^* (*p* = 0.0032), *Il-10^−/−^* × *Tdag8*^−/−^ (*p* < 0.001), and *Il-10*^−/−^ × *Ogr1*^−/−^ × *Tdag8*^−/−^ (*p* = 0.0059) mice.

### 2.3. The Absence of Ogr1 Reduces Pro-Inflammatory Cytokine Expression in Tdag8-Deficient mice in the Model of Spontaneous Colitis

Transcriptional analysis showed a trend towards a decreased expression of *Tnf* and *Il-6* in *Il-10*^−/−^ × *Ogr1*^−/−^ mice when compared with *Il-10*^−/−^ mice (Figure 5A,B). In contrast, *Il-10*^−/−^ × *Tdag8*^−/−^ mice showed a significantly increased *Il-6* expression compared with *Il-10*^−/−^ mice (*p* = 0.0096, Figure 5B). The corresponding increase in *Tnf* expression was not significant (*p* = 0.1657, Figure 5A). Interestingly, *Il-10*^−/−^ × *Ogr1*^−/−^ × *Tdag8*^−/−^ mice showed a trend towards a decreased *Tnf* expression compared with *Il-10*^−/−^ × *Tdag8*^−/−^ mice (*p* = 0.0719) and a significant reduction in *Il-6* expression levels (*p* = 0.0061), suggesting a decreased inflammation in the triple KO mice.

### 2.4. Ogr1-Deficient Mice Present Decreased Pro-Inflammatory Cell Populations in the Colon Even in the Absence of Tdag8

Histological analysis revealed an increased influx of lymphocytes in the colon of *Il-10*^−/−^ and *Il-10*^−/−^ × *Tdag8*^−/−^ mice compared with *Il-10*^−/−^ × *Ogr1*^−/−^ and *Il-10*^−/−^ × *Ogr1*^−/−^ × *Tdag8*^−/−^ animals (Figure 4A,B). Subsequently, we focused on the presence of pro-inflammatory T-cell populations in the colon using FACS analysis. The number of TNF^+^ CD4^+^ T-cells (Figure 6A and Appendix A) were significantly decreased in colon samples from *Ogr1*-deficient mice compared with *Il-10*^−/−^ mice (*p* = 0.0085), whereas the number of these cells was unaffected in the absence of *Tdag8* (Figure 6A). *Il-10*^−/−^ × *Ogr1*^−/−^ × *Tdag8*^−/−^ mice showed a decreased number of TNF^+^ CD4^+^ T-cells compared with *Il-10*^−/−^ mice (*p* = 0.0330), confirming the data obtained by qPCR analysis. A similar picture emerged when measuring the number of IFN^+^ CD4^+^ T-cells (Figure 6B). *Il-10*^−/−^ × *Tdag8*^−/−^ mice showed a similar number of IFN^+^ CD4^+^ T-cells to *Il-10*^−/−^ mice and a trend toward an increased number compared with *Il-10*^−/−^ × *Ogr1*^−/−^ (*p* = 0.0726) and *Il-10*^−/−^ × *Ogr1*^−/−^ × *Tdag8*^−/−^ mice (*p* = 0.0703, Figure 6B). The *IL-23* pathway plays a key role in IBD pathogenesis through the promotion of a pathological Th17 response, which in turn has been linked to *Tdag8* function. Consequently, we assessed IL-23 in γδT-cells. The number of IL-23^+^ γδTCR^+^ T-cells was significantly increased in colon samples from *Il-10*^−/−^ × *Tdag8*^−/−^ mice compared with *Il-10*^−/−^ animals (*p* = 0.0138, Figure 6C and Appendix A) and deficiency of *Ogr1* (*Il-10*^−/−^ × *Ogr1*^−/−^ × *Tdag8*^−/−^ mice) reduced this level back to the corresponding control (*Il-10*^−/−^ × *Tdag8*^−/−^ mice, *p* = 0.0038, Figure 6C). The amount of IL-23^+^ γδTCR^+^ T-cells was not increased in *Il-10*^−/−^ animals compared with *Il-10*^−/−^ × *Ogr1*^−/−^ and *Il-10*^−/−^ × *Ogr1*^−/−^ × *Tdag8*^−/−^ mice. The number of γδTCR^+^ B-cells, Ly6C^+^ CD11b^+^ monocytes, and F4/80^+^ CD64^+^ macrophages remained unchanged in the colon samples of all genotypes. Subsequently, we analyzed the FoxP3^+^ CD4^+^ Treg cell population in the colon (Figure 6D and Appendix A). This population was significantly increased in colon samples from *Il-10*^−/−^ × *Ogr1*^−/−^ mice compared with *Il-10*^−/−^, and *Il-10*^−/−^ × *Tdag8*^−/−^ mice (*p* < 0.0086, 0.0164, respectively, Figure 6D). We also analyzed the presence of F4/80^+^ CD64^+^ macrophages in the colon (Figure 4E and Appendix A). These cells were significantly more prominent in colon samples from *Il-10*^−/−^ × *Tdag8*^−/−^ mice compared with *Il-10*^−/−^, *Il-10*^−/−^ × *Ogr1*^−/−^, and *Il-10*^−/−^ × *Ogr1*^−/−^ × *Tdag8*^−/−^ mice (*p* < 0.0001 each, Figure 6E). Further, we determined the presence of the CD11b^+^ Ly6G^+^ neutrophil population in the colon (Figure 6F and Appendix A). The number of neutrophils was significantly decreased in colon samples from *Il-10*^−/−^ × *Ogr1*^−/−^ mice compared with *Il-10*^−/−^ and *Il-10*^−/−^ × *Tdag8*^−/−^ mice (*p* = 0.0099 and 0.0375, respectively).

Given the high number of macrophages in *Il10*^−/−^ × *Tdag8*^−/−^ mice in the colon, we analyzed this cell population in spleen samples by FACS (Figure 7A and Appendix A). The number of F4/80^+^ CD64^+^ macrophages was significantly increased in *Il-10*^−/−^ × *Tdag8*^−/−^ mice compared with *Il-10*^−/−^, *Il-10*^−/−^ × *Ogr1*^−/−^, and *Il-10*^−/−^ × *Ogr1*^−/−^ × *Tdag8*^−/−^ animals (*p* < 0.0001 each, Figure 7A). The majority of F4/80^+^ CD64^+^ macrophages belong to the CD206^+^ CD163^+^ M2c subtype and remained unchanged in all genotypes (Appendix A). Since pro-inflammatory IL-17 plays a key role in IBD pathogenesis through the recruitment of monocytes and neutrophils, we also assessed the number of IL-17A^+^ γδTCR^+^ T cells (Figure 7B and Appendix A). The number of IL-17A^+^ cells was significantly increased in spleen samples from *Il-10*^−/−^ × *Tdag8*^−/−^ mice compared with *Il-10*^−/−^, *Il-10*^−/−^ × *Ogr1*^−/−^, and *Il-10*^−/−^ × *Ogr1*^−/−^ × *Tdag8*^−/−^ mice (*p* < 0.0001 each, Figure 7B).

## 3. Discussion

In the present study, we investigated the role of anti-inflammatory TDAG8 and pro-inflammatory OGR1 proton-sensing receptors using well-established murine models of acute and spontaneous colitis. It is interesting to note that in the IL10 deficiency model (spontaneous colitis) the pro-inflammatory effects of TDAG8 deficiency became clearly visible and highly significant. We can therefore state that TDAG8 deficiency exacerbates colonic inflammation already triggered by a lack of IL10. Exacerbation was particularly prominent in overall disease scores and when measuring IL6 and F4/80^+^ CD64^+^ macrophages as well as IL23^+^ γδ T cells in the colon. Spleen weight also reflected exacerbation and macrophages and IL17^+^ T cells were found increased in spleen. These data are in agreement with the notion that in *Il10*-receptor-deficiency colonic inflammation is strongly driven by inflammatory macrophages which produce IL6 [40] and thus support increases in IL23 and subsequently Th17 pathology [41,42,43]. Importantly, absence of *Ogr1* led to a significant amelioration of inflammation both in the DSS model, as described before [24], and in the *Il10* deficiency model, even when *Tdag8* was knocked out in addition to IL10. In the latter case, *Ogr1* deletion led to significantly decreased levels of *Il-6* in the whole colon, and a decreased percentage of IL-23^+^ γδ T cells and F4/80^+^ macrophages in the colon and a decrease in both F4/80^+^ and IL-17A^+^ splenocytes, compared with IL10^−/−^ × TDAG8^−/−^ littermates. These results demonstrate that deletion of the pro-inflammatory receptor *Ogr1* can largely balance out the absence of anti-inflammatory *Il-10* and *Tdag8*.

There is now strong evidence that IBD pathogenesis is driven by IL-23 and its receptor IL-23R which promote a pathological Th17 response [44]. The pH-sensing receptor TDAG8 likely plays a role in this sequence of events. Lactic acidosis stimulates the production of IL-23 by mononuclear phagocytes, promoting the expected Th17 profile [45]. *Il-17a* was increased in *Tdag8*-deficient mice in a model of myocardial infarction [46]. It was further shown that TDAG8 promotes Th17 cell pathogenicity [47] and is required for the maintenance of lysosomal function, in particular for autophagy and pathogen defense [48]. An impairment of the function of TDAG8 also enhances antigen presentation of dendritic cells to T-cells, thereby promoting inflammation [49]. TDAG8 is also required for lysosome homeostasis and protein degradation, which play an important role in phagocytosis [49]. In the colon, TDAG8 deficiency or the coding variant I231L, an inflammatory disease-associated variant, promote inflammation and fibrosis by impairing the differentiation of CD4^+^ T-cells to Th17- and Th22-cells, and enhancing pro-inflammatory cytokines while reducing anti-inflammatory cytokines [12,13,49]. A genetic loss of function in the coding region of TDAG8 was described to affect lysosomal pH, thereby linking lysosomal dysfunction with increased risk of developing colitis [48].

Conversely, OGR1 is induced by hypoxia via hypoxia-inducible factor (HIF)1α and TNF, which are associated with inflammation [23,29]. OGR1-KO mice showed a reduction in the expansion of Th17 and Th1 cells in a murine model of autoimmune encephalomyelitis [50]. OGR1 signaling in immune and stromal cells appears required for production of IL6 and IL23 in inflammation, and the protective effect of *Ogr1* deficiency appears to involve the suppression of the IL-23 pathway and therefore the Th17 response in *Il-10^−/−^* × *Tdag8^−/−^* × *Ogr1^−/−^* mice in our study. Of note, OGR1 may also negatively regulate IL-10 production from differentiating CD4^+^ cells as activation of OGR1 with ogerin resulted in suppression of IL-10 secretion by T cells [51]. 

Clearly, IL-10 and TDAG8 act on complementary routes to inhibit inflammation, otherwise the exacerbation of disease in *Il-10^−/−^* × *Tdag8^−/−^* animals observed here would not be measured. Our data as well as other reports suggest that macrophages are key in eliciting colitis in IL10 deficiency [40,41,42,43,52]. It will be interesting to define in greater detail the interplay of these complementary pathways in this cell type. In this context it is worth noting that activation of TDAG8 was described to increase IL-10 secretion in macrophages [53]. 

This study has several limitations. The DSS-induced colitis model only reflects the immunological situation during acute inflammation, while Il-10-deficient mice are useful in determining long-term effects of ongoing inflammatory initiated spontaneously. In comparison, defining features of IBD include severe and long-lasting mucosal tissue destruction interrupted by gradually shorter remission phases. Further studies are needed to study the effects of the double KO in an animal model that recapitulates the relapsing-remitting pattern of IBD.

Regarding potential new therapeutic approaches in IBD, our data suggest that agonism of TDAG8 may be particularly useful in situations where IL10 signaling is diminished. OGR1 inhibition should be investigated as a therapeutic option generally in IBD, as this pH-sensing receptor appears to sustain inflammation acting on pro-inflammatory pathways that are sensitive to inhibition by IL-10. Inhibition of OGR1 could be of special interest to patients carrying mutations in the CD susceptibility gene TDAG8.

## 4. Materials and Methods

### 4.1. Animals

All animal experiments were performed according to the ARRIVE criteria. The generation, breeding, and genotyping of C57BL/6 B6-Gpr68 < tm1Dgen > (*Ogr1^−/−^*) and C57BL/6 B6-Gpr65 < tm1Dgen > (*Tdag8^−/−^*) mice, initially obtained from Deltagen, Inc., San Mateo, CA, has been described previously [13]. The double KO mice B6-Gpr65 < tm1Dgen> × B6-Gpr68 < tm1Dgen > (*Ogr1^−/−^* × *Tdag8^−/−^*) were bred from the aforementioned mice and genotyped. The animal experiment protocol was approved by the Veterinary Authority of the canton of Zurich (registration number ZH 113/2021). Littermates were used in all experiments. The animals were co-housed wherever possible and bedding was exchanged among the cages to minimize potential effects of microbiota variation. All the animals were housed in a specific pathogen-free facility. The animals were kept in type II long clear-transparent individually ventilated cages (IVCs, 365 mm × 207 mm × 140 mm, Allentown, NJ, USA) with autoclaved dust-free bedding and tissue papers as nesting material. They were fed a pelleted and extruded mouse diet (R/M–H Extrudat, ssniff Spezialdiäten, Soest, Germany) ad libitum. The light/dark cycle in the room was provided through natural daylight (sunrise: 07:00 h, sunset: 18:00 h). The mice were weighed at 10:00 h every morning. The temperature was set to 21 ± 1 °C, with a relative humidity of 55 ± 5% and 75 complete changes of filtered air per hour (filter: Megalam MD H14, Camfil, Zug, Switzerland).

### 4.2. DSS-Induced Acute Colitis

Acute colitis was induced in WT, *Ogr1^−/−^*, *Tdag8^−/−^*, and *Ogr1^−/−^* × *Tdag8^−/−^* mice by the addition of 2.5% DSS (36–50 kDa, MP Biomedicals, Santa Ana, CA, USA) in the drinking water for eight consecutive days. Mouse body weight and clinical phenotype were assessed daily.

### 4.3. Spontaneous Il-10 Deficient Colitis Model 

B6.129P2-Il10 < tm1Cgn > /J (*Il-10^−/−^*) mice were crossed with the above-mentioned strains to generate mice susceptible for spontaneous colitis (*Il-10^−/−^*, *Ogr1^−/−^* × *Il-10^−/−^*, *Tdag8^−/−^* × *Il-10^−/−^*, and *Ogr1^−/−^* × *Tdag8^−/−^* × *Il-10^−/−^*, *n* = 17, 13, 8, 9, respectively). Mice were observed for 291 days during which mouse body weight and clinical phenotype were assessed. 

### 4.4. Assessment of Colonoscopy and Histological Score in mice 

Prior to endoscopic assessment, the animals were anesthetized intraperitoneally with a mixture of 90–120 mg ketamine (Narketan 10%, Vétoquinol AG, Bern, Switzerland) and 8 mg xylazine (Rompun 2%, Bayer, Zürich, Switzerland) per kg body weight, and examined with the Tele Pack Pal 20043020 (Karl Storz Endoskope, Tuttlingen, Germany). Mice were scored with a MEICS [54]. For the assessment of the histological scores, 1 cm of the distal third of the colon was removed, fixed in paraformaldehyde solution (4% in PBS, Santa Cruz Biotechnology, Santa Cruz, CA, USA), embedded, stained with hematoxylin and eosin (HE), and scored as described [55,56]. 

### 4.5. Ribonucleic Acid (RNA) Isolation, Complementary DNA (cDNA) Synthesis, and qPCR

Total RNA was isolated from colon and ileum using the Maxwell RSC simplyRNA tissue kit (Promega, Madison, WI, USA, AS1340). For all samples, lysis buffer from the kit was added to snap frozen resections, and samples were shredded in M tubes (Miltenyi Biotec, Bergisch Gladbach, Germany) using a gentleMACS tissue homogenizer (Miltenyi Biotec). RNA concentration was determined by absorbance at 260 and 280 nm with a NanoDrop (Thermo Fisher Scientific, Waltham, MA, USA). cDNA synthesis was performed using a High-Capacity cDNA Reverse Transcription Kit (Applied Biosystems, Foster City, CA, USA) following the manufacturer’s instructions. qPCR was performed using the TaqMan Fast Universal Master Mix (Applied Biosystems) on a QuantStudio™ 6 Flex Real-Time PCR System and results were analyzed with the SDS software (Applied Biosystems). For each sample, triplicates were measured, and glyceraldehyde-3-phosphate dehydrogenase (*Gapdh*) was used as endogenous control. Results were analyzed using the ∆∆CT method. The following gene expression assays for mice were used: *Tnf* Mm00443259_g1 (Thermo Fischer Scientific), *Il-6* Mm00446190_m1 (Applied Biosystems), and *Gapdh* 4352339E.

### 4.6. FACS

For single-cell FACS analysis, lamina propria lymphocytes and splenocytes were isolated. Surface antigens were stained with a mix of antibodies including a viability marker (Appendix A) and incubated at 4 °C for 20 min. After washing with phosphate-buffered solution (PBS) and centrifugation, all the samples were fixed. For fixation, BD Cytofix/Cytoperm (554722) and BD Perm/Wash (554723) were used following the manufacturer’s instructions. The pellet was then resuspended in PBS. Intracellular antigens were stained with a mix of antibodies (Appendix A). Data were acquired on a FACS LSR II Fortessa 4L (BD) and analyzed with the FlowJo software (version 10.2).

### 4.7. Lamina Propria Lymphocytes Isolation

Following the manufacturer’s instructions, Hanks’ balanced salt solution (HBSS) was prepared from the solid salts (Sigma Aldrich, Burlington, MA, USA, Hanks’ Balanced Salts, H2387-10X1L). The enzyme mix for digestion was prepared as follows: 0.4 mg/mL collagenase type IV (Gibco, Billings, MT, 17104-019) and 0.6 mg/mL dispase (Gibco, 17105-041) were added to 10% fetal calf serum (FCS) low Ig (PAN Biotech, Aidenbach, Germany, P30-2802) in HBSS colorless (Sigma Aldrich, H8264-1L). Cells were isolated from 2 cm of the distal colon per sample. The samples were first transferred in 2 mM EDTA (Sigma Aldrich, 38057-1EA) in HBSS for 15 min at 37 °C while shaking. The supernatant was removed, and the samples were incubated in 2 mM EDTA in HBSS for another 30 min. The supernatant was removed again, and the samples were incubated in warm HBSS before transferring into the previously prepared enzyme mix for digestion and incubated for 20 min at 37 °C while shaking. After incubation, the samples were sheared with the help of a syringe and 18 G needle and then passed through a 70 µm cell strainer. The supernatant was removed after centrifugation, and the cells were resuspended in PBS before staining and fixation.

### 4.8. Splenocyte Isolation

Spleen samples were cut up, passed through a 70 µm cell strainer with PBS, and centrifuged. The formed pellet was then lysed with 3 mL ACK buffer (NH_4_Cl 0.15 M, KHCO_3_ 0.01 M, EDTA-Na_2_ 0.001 M) for 3 min at room temperature. After resuspension with PBS, the solution was filtered with a 70 µm cell strainer and centrifuged again. The pellet was then resuspended in PBS before staining and fixing the cells.

### 4.9. Compensation Controls

Two different bead kits were used for compensation controls: ArC Amine reactive compensation bead kit (Thermo Fisher Scientific, A10346) for the cell viability marker. For the remaining antibodies, the BD CompBeads kit anti-rat and anti-hamster (Becton Dickinson Pharmingen Biosciences, San Diego, CA, USA, 552845), and the BD CompBeads kit anti-mouse (Becton Dickinson Pharmingen Biosciences, 552843) were used. All the compensation controls were prepared following the manufacturer’s instructions.

### 4.10. Statistical Analysis

Statistical analyses were performed using GraphPad Prism 8 [GraphPad Software], version 9.5.1. (733), with analysis of variance [ANOVA]. Normal distribution was analyzed using the Shapiro–Wilk test, followed by Tukey post hoc test (Figure 3A,D, Figure 4B, Figure 5A, Figure 6B,C,E,F, and Figure 7A,B) or non-parametric Kruskal–Wallis test followed by multiple comparison test with Benjamini, Krieger, and Yekutieli correction (Figure 1A,B,D–F, Figure 2B, Figure 3B,E,F, Figure 5B, and Figure 6A,D). Differences were considered significant at a *p*-value < 0.05, as indicated in the figures. Results are presented as mean ± standard deviation (SD) or ± standard error of the mean (SEM), as indicated in the figure legends.

## Figures and Tables

**Figure 1 ijms-24-14855-f001:**
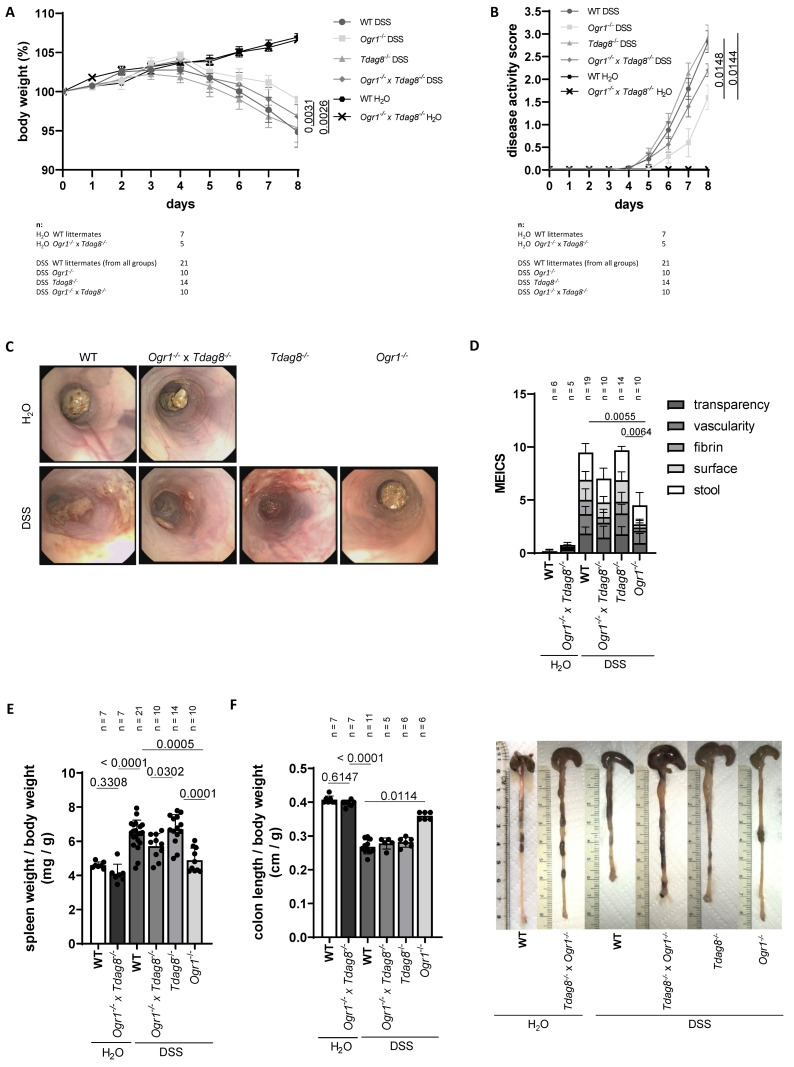
The absence of OGR1 reduces inflammation upon DSS-induced acute colitis. (**A**) Body weight, ±SEM. (**B**) Clinical disease activity score ± SEM. (**C**) Coloscopy, representative images of each group. (**D**) MEICS, ±SD. (**E**) Spleen weight/body weight ± SEM. (**F**) Colon length/body weight, ±SD, and exemplary pictures of colons from each group.

**Figure 2 ijms-24-14855-f002:**
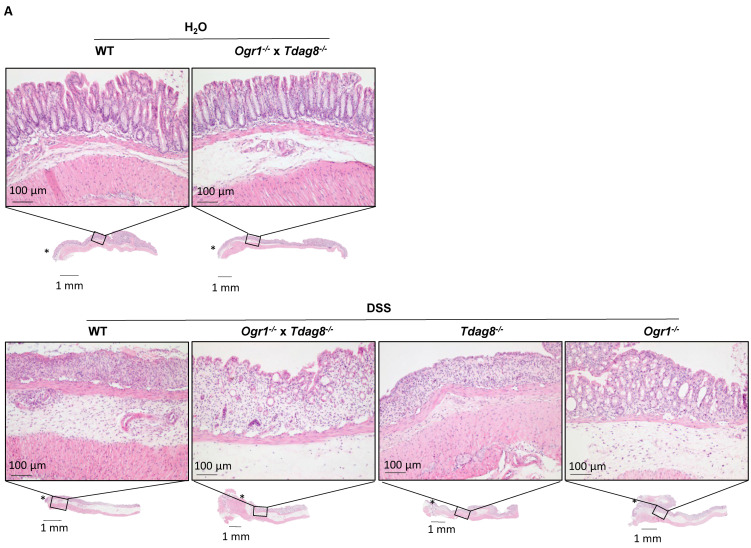
The absence of OGR1 reduces the histological score upon DSS-induced acute colitis. (**A**) Exemplary microscopic pictures of HE stained colons. (**B**) Histological score, ±SD. One-way ANOVA, *p*-values and *n* as indicated, * *p* < 0.05.

**Figure 3 ijms-24-14855-f003:**
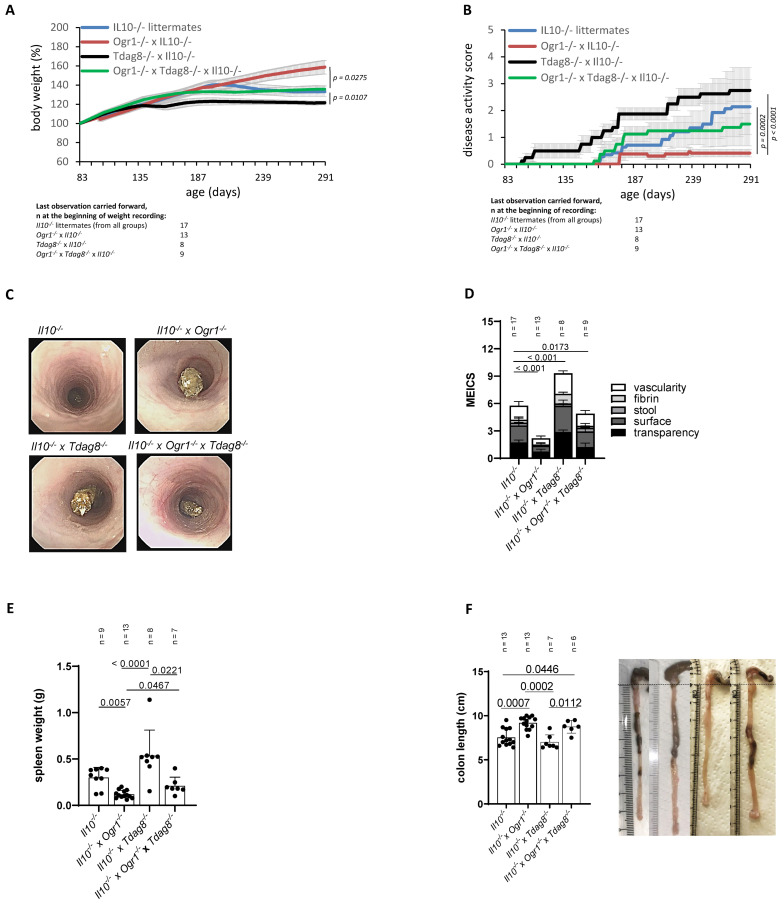
The absence of TDAG8 aggravates inflammation in the spontaneous colitis model. (**A**) Body weight ± SEM. (**B**) Clinical disease activity score ± SEM. (**C**) Coloscopy, representative images of each group. (**D**) MEICS compared between groups, ±SD. (**E**) Spleen weight, ±SD. (**F**) Colon length, ±SD, and representative images of colons from each group.

**Figure 4 ijms-24-14855-f004:**
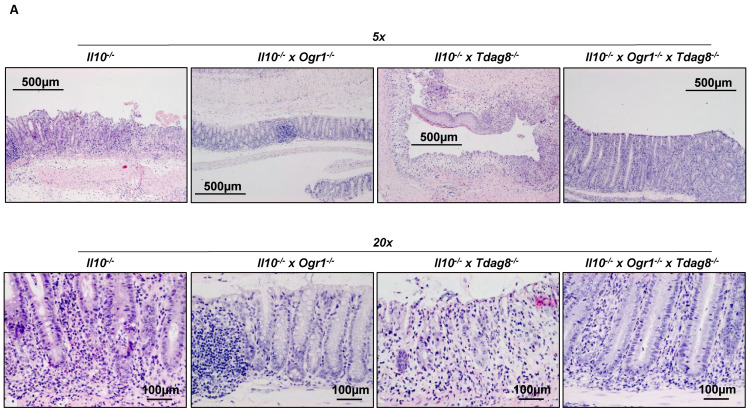
The absence of TDAG8 aggravates the histological score in the spontaneous colitis model. (**A**) Representative images of HE stained colons from each group. (**B**) Histological score, ±SD. One-way ANOVA, *p*-values and *n* as indicated.

**Figure 5 ijms-24-14855-f005:**
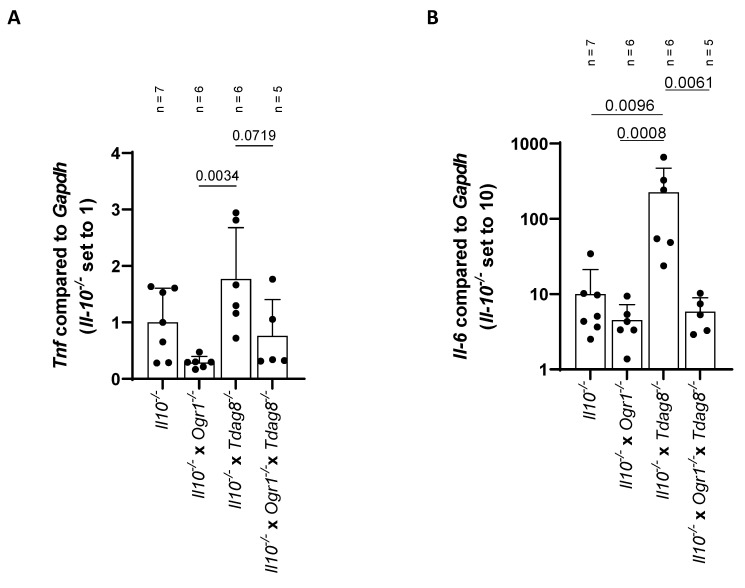
The absence of TDAG8 aggravates inflammation in the chronic colitis model. qPCR results for *Tnf* and *Il-6* in the spontaneous colitis model, ±SD, ordinary one-way ANOVA, with *p*-values as indicated. (**A**) *Tnf* compared with *Gapdh* with *Il-10^−/−^* set to 1. (**B**) *Il-6* compared with *Gapdh* with *Il-10^−/−^* set to 10 (logarithmic y-axis).

**Figure 6 ijms-24-14855-f006:**
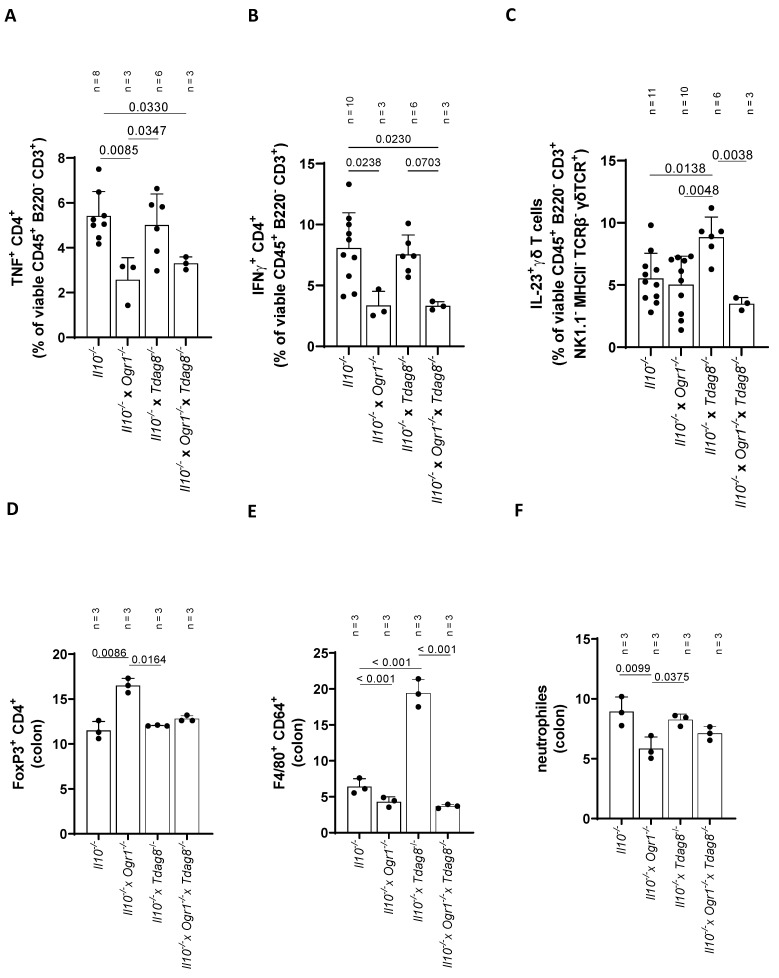
FACS results from colon samples of the spontaneous colitis mouse model. FACS analysis, ±SD, ordinary one-way ANOVA, with *p*-values as indicated. (**A**) TNF^+^ CD4^+^ cells isolated from colon. (**B**) Analysis of IFNγ^+^ CD4^+^ cells. (**C**) IL-23^+^ γδTCR T-cells. (**D**) FoxP3^+^ CD4^+^ Tregs. (**E**) F4/80^+^ CD64^+^. (**F**) Neutrophils.

**Figure 7 ijms-24-14855-f007:**
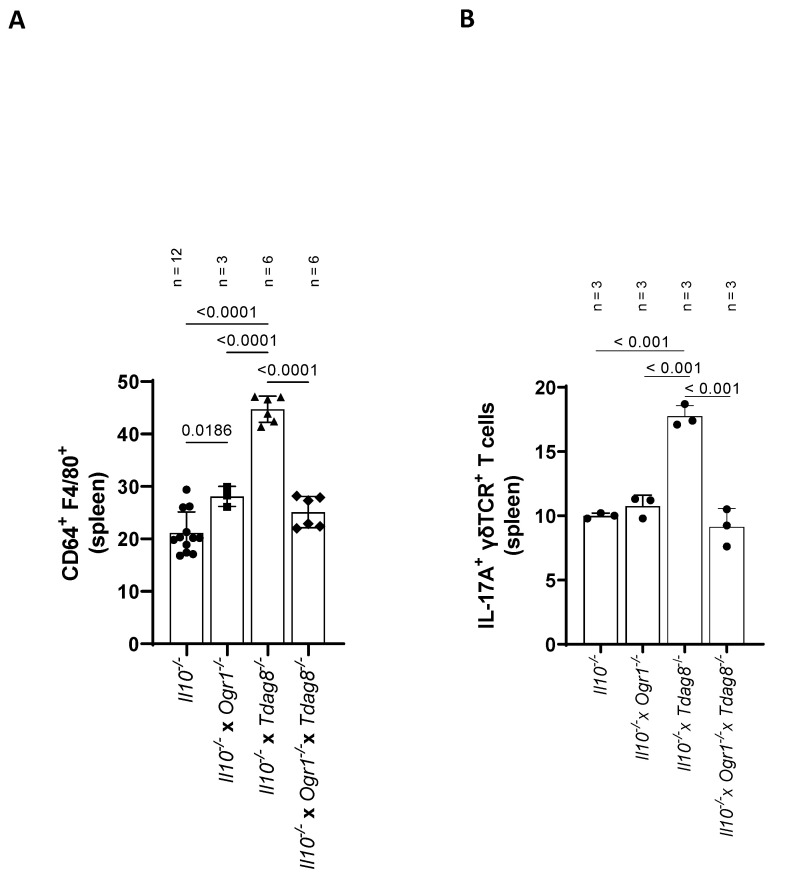
FACS results from spleen samples of the spontaneous colitis mouse model. FACS analysis, ±SD, ordinary one-way ANOVA, with *p*-values as indicated. (**A**) Manual gating of CD64^+^ F4/80^+^ macrophages isolated from spleen (% of viable CD45^+^ B220^−^ CD3^−^ CD11b^+^ Ly6G^+^ neutrophils excluded, CD11b^+^ Ly6C^+^ monocytes excluded, NK1.1^−^). (**B**) IL-17A^+^ (% of viable CD45^+^ B220^−^ CD3^+^ NK1.1^+^ γδTCR^+^ T-cells).

## Data Availability

The data underlying this article are available in a repository provided by the University of Zurich (the link to the repository still needs to be defined).

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
