# Peer review of "OGR1 (GPR68) and TDAG8 (GPR65) Have Antagonistic Effects in Models of Colonic Inflammation"

_ijms, 2023, doi:10.3390/ijms241914855_

Round 1
Reviewer 1 Report
Overall, this is a nicely done and well written publication. The study design is appropriate and apparently, the analyses were carefully performed. This manuscript shows rich and valuable content, which is within the journal’s scope. However, the study contains many shortness’s which need to be answered before publication.
My comments:
Line 16, 23, 61 etc. - From histological point of view there are only four kinds of tissues: epithelial, connective, muscular and nervous. Therefore, such terms as “colonic tissue”, “intestinal tissue”, “colon tumor tissue”, “mucosal tissue” etc. are not justified. The colon (intestine) is an organ but not a tissue.
Line 50 – the authors should ensure that they used term “expression” in relation to genes only.
Line 85 – please present your hypothesis correctly and the novelty of the study.
Line 89 – please avoid any outcomes in Introduction. They should be presented in conclusions.
Line 117 – Figure 1G and 2G are too small to make any judgement. Scale bars in 1G are barely seen and should be improved to fit one line.
Line 176 – Scale bars in Figure 2G are missing.
Line 336 – it is not clear how many animals in total were used. What were n=? for each group/experiment?
Line 369 – before embedding and further processing samples have to fixed first. What kind of fixative was used?
Line 424 – what kind of statistical test was applied to compare differences in means? What was a post-hoc test? Did the authors checked the normality assumption?
Line 329 – Please try to describe any limitations for this study.
Supplementary – Figures S1-S5 are missing.
Author Response
Reviewer 1
Line 16, 23, 61 etc. - From histological point of view there are only four kinds of tissues: epithelial, connective, muscular and nervous. Therefore, such terms as “colonic tissue”, “intestinal tissue”, “colon tumor tissue”, “mucosal tissue” etc. are not justified. The colon (intestine) is an organ but not a tissue.
- The word “tissue” has been replaced or deleted in all relevant places (except for the names of kits from companies).
Line 50 – the authors should ensure that they used term “expression” in relation to genes only.
- The term “expression” has been replaced or deleted in places where it was not used properly and now is only used in relation to genes.
Line 85 – please present your hypothesis correctly and the novelty of the study.
- The hypothesis and novelty of the study are now presented in the last paragraph of 1. Introduction: “This is the first study where the physiological consequences of the double deletion of OGR1 and TDAG8 have been investigated. The aim of this study was to better understand the link between the physiological role of both receptors, and assess which one has a predominant effect or is physiologically more relevant in the context of intestinal inflammation. We hypothesized that deletion of pro-inflammatory Ogr1 balances out the absence of anti-inflammatory Il-10, and Tdag8, and consequently OGR1 inhibition represents a potential therapeutic option in IBD, particularly for patients carrying mutations in the CD susceptibility gene TDAG8.”
Line 89 – please avoid any outcomes in Introduction. They should be presented in conclusions.
- The outcomes in the previous introduction have been deleted.
Line 117 – Figure 1G and 2G are too small to make any judgement. Scale bars in 1G are barely seen and should be improved to fit one line.
- The scale bars in 1G have been increased.
- The previous figures 1G and H have been moved into the new, enlarged figures 2 A and B.
- The previous figures 2G and H have been moved into the new, enlarged figures 4 A and B.
Line 176 – Scale bars in Figure 2G are missing.
- The scale bars in 2G have been added.
Line 336 – it is not clear how many animals in total were used. What were n=? for each group/experiment?
- The total n for the acute colitis and the spontaneous colitis models is now stated in 4.2 DSS-induced acute colitis and 4.3 Spontaneous Il-10 deficient colitis model.
- The n for each experiment is now stated within the figures 1A, B, D, E, F, 2B, 3A, B, D, E, F, 4B, 5A, B, 6A, B, C, D, E, F, 7A, B
- 1C, 2A, 3C, 4A contain only representative images.
Line 369 – before embedding and further processing samples have to fixed first. What kind of fixative was used?
- The fixing procedure is now mentioned in 4.4 Assessment of colonoscopy and histology score in mice: “fixed in paraformaldehyde solution (4% in PBS, Santa Cruz Biotechnology, USA)”
Line 424 – What kind of statistical test was applied to compare differences in means? What was a post-hoc test? Did the authors checked the normality assumption?
- The description of the statistical analysis has been updated in 4.10. Statistical analysis and now shows the requested information: “Statistical analyses were performed using GraphPad Prism 8 [GraphPad Software], with analysis of variance [ANOVA]. Normal distribution was analyzed using the Shapiro-Wilk test, followed by Tukey post hoc test (figure 3 A, D, 4 B, 5 A, 6 B, C, E, F, 7 A, B) or non-parametric Kruskal–Wallis test followed by multiple comparison test with Benjamini, Krieger and Yekutieli correction (figure 1 A, B, D, E, F, 2 B, 3 B, E, F, 5 B, 6 A, D).”
Line 329 – Please try to describe any limitations for this study.
- The limitations of the study are now described in in the second to last paragraph of 3. Discussion: “This study has several limitations. The DSS-induced colitis model only reflects the immunological situation during acute inflammation, while Il-10-deficient mice are useful in determining long-term effects of ongoing inflammatory initiated spontaneously. In comparison, defining features of IBD include severe and long-lasting mucosal tissue destruction interrupted by gradually shorter remission phases. Further studies are needed to study the effects of the double KO in an animal model that recapitulates the relapsing-remitting pattern of IBD.”
Supplementary – Figures S1-S5 are missing.
- The missing supplementary figures have been uploaded.
Reviewer 2 Report
The observation that TDAG8 deficiency exacerbates IL10 deficiency-induced colitis is interesting because many IBD patients develop recurrent colitis. Particularly, exacerbation was evident in the general cohort of induced inflammation mice, in which IL6 and F4/80+ CD64+ macrophages as well as IL23+ γδ T cells were measured in colonic tissue. Spleen weight also reflected exacerbation, as increased numbers of macrophages and IL17+ T cells were found in the spleen. These data are consistent with the view that in the case of IL-10 receptor deficiency, colitis is more severe, driven by inflammatory macrophages that produce IL6 and thus support the increase in IL23 and then Th17. In the Il10 deficiency model, even when Tdag8 was knocked down in addition to IL10. In the latter case, deletion of Ogr1 led to a significantly reduced level of Il-6 in the entire colonic connective tissue and a reduced percentage of IL-23+ γδ T cells and F4/80+ macrophages in the colon and a reduction in the number of F4/80+ splenocytes and IL-17A+ in the colon. compared with IL10-/- x TDAG8-/- associates. These results demonstrate that deletion of the pro-inflammatory receptor Ogr1 294 can largely compensate for the lack of anti-inflammatory Il-10 and Tdag8. There is now strong evidence that the pathogenesis of IBD is driven by IL-23 and its receptor. IL-23R track, promotes pathological Th17 response. The pH-sensing receptor TDAG8 likely plays a role in this sequence of events. Lactic acidosis is stimulated by the induction of IL-23 by mononuclear phagocytes, promoting the expected Th17 profile. Genetic loss of function in the TDAG8 coding region has been reported to affect lysosomal pH, thus linking lysosomal dysfunction with an increased risk of developing colitis. Conversely, OGR1 expression is induced by hypoxia via hypoxia-inducible factor (HIF)1α and TNF, which are associated with inflammation.
Interesting work, but the following information is missing:
1. what is the purpose of the work
2. please explain the mechanism of influence of the GR1 and TDAG8 genes, as these proteins are activated by acidic extracellular pH
Author Response
Reviewer 2
1. What is the purpose of the work
- Reviewer 1 had a similar query. The purpose of the study is now stated in the last paragraph of 1. Introduction: “This is the first study where the physiological consequences of the double deletion of OGR1 and TDAG8 have been investigated. The aim of this study was to better understand the link between the physiological role of both receptors, and assess which one has a predominant effect or is physiologically more relevant in the context of intestinal inflammation. We hypothesized that deletion of pro-inflammatory Ogr1 balances out the absence of anti-inflammatory Il-10, and Tdag8, and consequently OGR1 inhibition represents a potential therapeutic option in IBD, particularly for patients carrying mutations in the CD susceptibility gene TDAG8.”
2. Please explain the mechanism of influence of the OGR1 and TDAG8 genes, as these proteins are activated by acidic extracellular pH.
- The questions can only be answered inadequately at the moment. Several second messengers of OGR1 and TDAG8 have been identified and are described in 1. Introduction (second and third paragraph, respectively). Briefly, TDAG8 activates the Gαs/cAMP and Gα12/13/Rho pathways to reduce the expression of pro-inflammatory cytokines (e.g., IL-1β). Conversely, OGR1 couples through Gq11 proteins leading to activation of the PLC/IP/Ca2+/ERK pathway, as well as the Gα12/13/Rho signaling pathway, which ultimately leads to the increased
expression of pro-inflammatory mediators. To date, there are no data on a potential mutual influence of pathways activated by both receptors and their opposite effects on inflammation seem to arise from the activation of different transcription factors and cellular processes whose contribution ultimately modulate inflammation under acidic conditions.
We believe this is a crucial question that needed to be addressed, and consequently we have updated the 1. Introduction: “To date, there are no data on a potential mutual influence of the pathways activated by both receptors, and their opposite effects on inflammation seem to arise from the activation of different transcription factors and cellular processes. Thus, OGR1 appears to block autophagy and induce the activation of the endoplasmic reticulum stress transcription factor X-box-binding protein-1 through the IRE1α-JNK signalling pathway [Maeyashiki et al. Sci Rep. 2020]. Following activation by an acidic environment, different downstream signals from both receptors seem to trigger independent molecular mechanisms whose contribution ultimately modulate inflammation under acidic conditions.”
Round 2
Reviewer 1 Report
The authors reasonably responded to my concerns.